# Beneficial Biological Activities of *Cinnamomum osmophloeum* and Its Potential Use in the Alleviation of Oral Mucositis: A Systematic Review

**DOI:** 10.3390/biomedicines8010003

**Published:** 2020-01-01

**Authors:** Abu Bakar, Pin-Chuan Yao, Valendriyani Ningrum, Cheng-Tzu Liu, Shih-Chieh Lee

**Affiliations:** 1PhD Program of Biotechnology and Industry, Da-Yeh University, Dacun, Changhua 51591, Taiwan; abuba.mmed@gmail.com; 2School of Dentistry, Baiturrahmah University, Kuranji, Padang 25155, Indonesia; 3Department of Material Science and Engineering, Da-Yeh University, Dacun, Changhua 51591, Taiwan; 4School of Nutrition, Chung Shan Medical University, Taichung 40201, Taiwan; 5Department of Nutrition, Chung Shan Medical University Hospital, Taichung 40201, Taiwan; 6Department of Food Science and Biotechnology, Da-Yeh University, Dacun, Changhua 51591, Taiwan

**Keywords:** *C. osmophloeum*, biological activities, oral mucositis, cinnamaldehyde, linalool

## Abstract

The aim of this review was to provide an updated overview of studies on the medical-biological activities of *Cinnamomum osmophloeum* (*C. osmophloeum*) in vitro and in vivo and the potential therapeutic use of natural agents prepared from this plant for the alleviation of oral mucositis (OM). Reported articles were collected using web search engine tools. The systematic review was organized according to the preferred reporting items for reviews and meta-analyses (PRISMA) statement. Additional sources were identified through cross-referencing to identify the potential use of *C. osmophloeum* in the alleviation of OM. The results disclosed that *C. osmophloeum* is comprised of bioactive ingredients that could act diversely as a reagent in anti-inflammation, antibacterial, antioxidant, anti-hyperglycemic, antidyslipidemia, anti-cancer, renal disease therapy and anti-hyperuricemia capacities. Recent studies revealed that the overall effects on anti-inflammation, wound repair, and the antibacterial and antioxidant activities of its constituents would act as a potential remedy for oral mucositis. Up-to-date in vitro and in vivo studies on the medical-biological activities of *C. osmophloeum* suggested that *C. osmophloeum* and its constituents could be promising remedies as adjuvants in OM therapy and warrant further investigation.

## 1. Introduction

Cinnamon plants (*Cinnamomum* spp.) are of the genus Lauraceae, native to South and Southeast Asia, and are generally used as food flavors and traditional medicinal plants. *Cinnamomum osmophloeum*, commonly known as indigenous cinnamon or pseudocinnamon, is endemic to Taiwan’s natural hardwood forests [1]. Major components of the essential oils extracted from *C. osmophloeum* leaves explored by high-performance liquid chromatography (HPLC) are as follows: α-pinene, camphene, benzaldehyde, β-pinene, 3-pheayl pionaldehyde, *cis*-cinnamaldehyde, *trans*-cinnamaldehyde, isobornylacetate, eugenol, and cinnamil acetate [2]. The essential oils extracted from *C. osmophloeum* leaves comprise 101 volatile compounds, as identified by GC/MS analysis, including monoterpenoids, sesquiterpenoids, alcohols, phenols, aldehydes, ketones, esters, acids, and other miscellaneous compounds. It was demonstrated that the linalool chemotypes present in *C. osmophloeum* were as follows: linalool, trans-cinnamyl acetate, camphor, cinnamaldehyde, 3-phenyl-2-propenal, caryophyllene, coumarin, bornyl acetate, limonene, α-(+)-pinene, estragole, and caryophyllene oxide [3]. In several studies (both in vitro and in vivo), *C. osmophloeum* has been applied as an alternative natural therapy to treat certain compromised and uncompromised diseases [3,4,5,6,7].

Oral mucositis (OM) is known as the inflammation of oral mucosa, usually occurring as an adverse side-effect of chemotherapy and/or radiation therapy (radiotherapy), and is manifested as atrophy, swelling, erythema, and ulceration [8]. OM occurrence in the hospital might increase costs and deteriorate oral health quality of life [9,10,11]. Hence, oral care treatments, including nutritional care, pain control, oral cleansing, palliation of a dry mouth, bleeding handling, and medicinal interventions have been introduced to decrease the severity of OM after cancer therapy [12]. Patients receiving radiotherapy to head and neck areas are at a significant risk of developing oral mucositis. The risk is lower (less than 50% or little risk) in patients with prolonged chemotherapy, patients receiving surgery, and patients with radiotherapy to non-head and neck areas [10,13]. The underlying pathophysiology of OM is divided into five phases: (1) initiation, (2) primary damage response, (3) signaling and amplification, (4) ulceration (symptomatic phase), and (5) healing [14,15,16]. The first phase (initiation stage) happens after exposure to radiotherapy or chemotherapy. It consists of two events: DNA breakdown and the generation of reactive oxygen species (ROS). DNA strand breakdowns lead to direct injury and death of the cells, and reactive oxygen species play a role as key initiators and mediators of downstream biological events. During the second phase (primary damage response), activator transduction pathways are stimulated by the DNA breaks strand, which can lead to the activation of several transcription factors, including p53 and nuclear factor kappa-B (NF-κB). NF-κB works as a controller for the expression of a broad range of genes, and produces a series of mediators, including pro-inflammatory cytokines and both pro- and anti-apoptotic cellular changes. During phase III (signal amplification stage), pro-inflammatory cytokines deliver a positive reaction to enhance and accelerate the process of wound healing. During phase IV (ulceration phase, also called symptomatic phase), it is common for the mucosal surface to become re-infected with bacteria. Bacterial invasion stimulates macrophage accumulation to conceal additional amounts of pro-inflammatory cytokines. During phase V (healing stage), signals from the connective tissue to the bordering epithelium can activate the migration, propagation, and differentiation of cells, resulting in healed mucosa. A number of biomaterials explored for OM therapy perform their principal mechanisms linked to pathophysiology by depressing pro-inflammatory cytokines. In addition to anti-inflammation, antioxidant, antifungal, antibacterial, and immunomodulator mechanisms of action have been reported [17]. As one of the encouraging biomaterials, *C. osmophloeum* might be a potential alleviator of OM.

Previous studies on cinnamon have been limited to *C. osmophloeum* [1]. However, different species of cinnamon might possess diverse beneficial biological activities aiming at various medical therapies. Therefore, we have provided an updated extensive overview on studies of the medical-biological activities of *C. osmophloeum* and its principal ingredients, both in vitro and in vivo. The potential use of this natural component in oral mucositis therapy is also discussed.

## 2. Materials and Methods

A computerized search for the reported works was performed through PubMed^®^, Scopus, ScienceDirect, and Web of Science databases by using the key words “*cinnamon osmophloeum*”, “*indigenous cinnamon*”, and “*pseudocinnamon*”. A systematic review was conducted following PRISMA. In order to identify the potential use of *C. osmophloeum* in the alleviation of OM, additional sources were identified through cross-referencing.

In addition, previously published review articles were searched to gain additional information. In order to identify the potential use of *C. osmophloeum* in the alleviation of OM induced by radiotherapy and/or chemotherapy, a second computerized search using PubMed^®^, Scopus, ScienceDirect, and Web of Science databases was carried out by utilizing keywords with *C. osmophloeum* constituents that possess anti-inflammation, antioxidant, and antibacterial activities with OM, including “*cinnamaldehyde* + *oral mucositis*”, “*trans-cinnamaldehyde* + *oral mucositis*”, “*kaempferitin* + *oral mucositis*”, “*kaempferol* + *oral mucositis*”, “*cinnamic acid* + *oral mucositis*”, “*cinnamyl alcohol* + *oral mucositis*”, “*cinnamyl acetate* + *oral mucositis*”, “a*lloaromadendrene* + *oral mucositis*”, and “*linalool* + *oral mucositis*”.

## 3. Results and Discussion

### 3.1. Medical-Biological Activities of C. osmophloeum

A set of 51 articles related to *C. osmophloeum* was identified, in which 18 articles were excluded because they were duplicate papers (*n* = 2), were not related to the beneficial biological activities (*n* = 2), or were irrelevant articles (*n* = 14). Irrelevant articles involved studies not related to medical uses, studies related to mushroom planting in *C. osmophloeum*, and review article(s) (Figure 1). Specific studies (2 to 4) discussed the beneficial biological activities [3,18,19,20,21,22] and suggested more than one chemical compound as the active content. Some beneficial biological activities have been summarized in Table 1 and consist of anti-inflammatory activities [20,22,23,24,25,26,27,28,29,30], wound repair activities [20], antibacterial activities [31,32], antifungal activities [33,34], antioxidant activities [18,20,35,36,37], anti-hyperglycemic activities [3,18,21], antidyslipidemia activities [38,39], anti-cancer and anti-tumor activities [22], renal disease therapy, and anti-hyperuricemia activities [2,4]. In addition, studies on the leaves, twigs, barks, heartwoods, and roots of *C. osmophloeum* were also reported.

It was conveyed that *C. osmophloeum* comprised six chemotypes; cinnamaldehyde, cinnamaldehyde/cinnamyl acetate, cinnamyl acetate, linalool, camphor, and mixed types (Table 2). The chemical structures of cinnamaldehyde, cinnamyl acetate, linalool, and camphor are given in Figure 2. Most of the studies showed that the dominant chemical contents of the cinnamaldehyde type were cinammaldehyde/*trans*-cinnamaldehyde [2,29,31,32,33,34,36,37,47,48], benzaldehyde [32,33,37,48], benzenepropanal [37,47,48,49], and cinnamyl acetate/*trans*-cinnamyl acetate [2,29,36]. The cinnamaldehyde/cinnamyl acetate type encompassed cinnamyl acetate, *trans*-cinnamaldehyde, and benzenepropanal [34,48,49]. The cinnamyl acetate type included cinnamyl acetate [29,34], 2-methylbenzofuran, and geranyl acetate [34]. The linalool type contained linalool [3,29,34,48] and cinnamaldehyde/*trans*-cinnamaldehyde [3,34]. The camphor type contained camphor/D-(+)-camphor and L-bornyl acetate [29,34,36,48] and limonene [34,48]. The mixed type comprised spathulenol [24], neral [32,33], L-bornyl acetate/bornyl acetate [29,30,34,41,48], 1,8-cineol [24,32,33], T-cadinol, and α-cadinol [29,34,41,48].

#### 3.1.1. Anti-Inflammatory and Wound Repair Activities

Numerous studies reported the anti-inflammatory effect of *C. osmophloeum* [24,25,26,28,29,30,40] and they have been applied to the medical treatment of animal organs, such as endotoxin-induced intestinal injury [27], pancreas protection [3], and lipopolysaccharide/D-galactosamine (LPS/D-GaIN)-induced acute hepatitis [6]. Some of these studies identified the specific constituents of *C. osmophloeum* that accounted for the anti-inflammatory effect [23,26,28,30,40]. For example, Kaempferitin, a constituent of *C. osmophloeum*, was reported to be an anti-inflammatory that could down-regulate the extracellular LDL-R (Low-Density Lipoprotein-Receptor), followed by ameliorating subsequent chronic inflammation-related diabetes mellitus [26]. Several constituents of *C. osmophloeum*, including trans-cinnamaldehyde, caryophyllene oxide, L-borneol, L-bornyl acetate, eugenol, β-caryophyllene, E-nerolidol, and cinnamyl, were demonstrated to be anti-inflammatory, which could suppress the synthesis of nitric oxide (NO) by LPS-stimulated macrophages through an identical mechanism [29,30]. Cinnamaldehyde was verified as the major anti-inflammation constituent whose activities were derived from the in vitro synthesis of LPS-stimulated macrophages [23].

Other constituents such as Kaempferol glycosides were associated with nitric oxide inhibitory activities [28]. In certain studies, linalool extracted from *C. osmophloeum* exhibited a protective effect on the pancreas by ameliorating pancreatic levels of interleukin (IL)-1β and nitric oxide [3], which play an important role in the prevention of LPS-induced inflammation in vivo [40].

*C. osmophloeum* has been used as a tyrosinase suppressor, which exhibited a substantial effect on wound repair. This phenomenon occurred through the inhibition of tyrosinase activity and reduced melanin content. Lee et al. carried out an animal experiment and explored the role of *C. osmophloeum* in wound repair and as an anti-oxidative agent. They evaluated a wound size assay, which revealed a decrease in the wound area by day 5 [20].

In light of the specific activities of the constituents of *C. osmophloeum*, it is noteworthy to mention that one specific and major constituent (flavanol glycosides) with outstanding properties was involved in inhibiting the production of NO and cytokines (TNF-a and IL-12), from LPS/IFNc-activated macrophages (inhibition of inflammatory activities) [25]. An additional study revealed the in vitro inhibitory effect of IL-1â and IL-6 production resulting from the 21 constituents of *C. osmophloeum* (cell-mediated immune response) observed [24]. Anti-inflammatory activity studies of *C. osmophloeum* have also been applied systemically in specific disease treatment [3,22,27,40]. Numerous mechanisms of inhibiting inflammation, including the compositional contents, mechanisms, anatomical parts, and promising investigations, have also been reported elsewhere [22,23,24,25,26,28,40,45].

#### 3.1.2. Antimicrobial Activities

The literature survey indicated that several pathogenic bacteria, including *Escherichia coli*, *Pseudomonas aeruginosa*, *Enterococcus faecalis*, *Staphylococcus aureus*, *Staphylococcus epidermidis*, methicillin-resistant *Staphylococcus aureus (MRSA)*, *Klebsiella pneumoniae*, *Salmonella* sp., *and Vibrio parahaemolyticus* have been used to investigate the antibacterial activity of *C. osmophloeum* [30]. For constituents of C. *osmophloeum*, the order of antibacterial activities was in the following sequence: cinnamaldehyde > cinnamic acid > cinnamyl alcohol > cinnamyl acetate. Cinnamaldehyde has been used as a bactericidal agent of *Legionella pneumophila* in vitro [31]. Two investigations confirmed that *C. osmophloeum* possesses antibacterial activities in both Gram-negative and Gram-positive bacteria [31,32]. Hence, further study should be carried out to examine animals infected by a specific group or both Gram-negative and Gram-positive bacteria.

*C. osmophloeum* also has antifungal properties in rot fungi, such as *Coriolus versicolor*, *Lenzites betulina*, *Pycnoporus coccineus*, *Trichaptum abietinum*, *Oligoporus lowe*, *Laetiporus sulphureus*, *Antrodia taxa*, *Fomitopsis pinicola*, and *Phaeolus schweinitzii* [33,34].

#### 3.1.3. Antioxidant Activities

As one of the biomaterials commonly used for traditional medical therapy, *C. osmophloeum* has been reported to be antioxidative, and its properties encompassing other medical-biological effects have been assessed systematically and independently by researchers worldwide [18,20,35,36,37,41].

For the analysis of anti-oxidant activities, both in vitro [18,20,35,37,41] and in vivo [35,36] investigations have been carried out. The investigations consisted of several studies: a 2,2-diphenyl-1-picrylhydrazyl (DPPH) scavenging activity assay [18,20,35], a superoxide radical scavenging assay (NBT Assay), reducing power, lipid peroxidation using mouse brain homogenates, the metal chelating ability, photochemiluminescence (PCL) [35], and an oxidative stress resistance assay [36]. Several constituents of *C. osmophloeum* comprising trans-cinnamaldehyde [36], alloaromadendrene [41], and kaempferol-7-O-rhamnoside [35] were proven to possess antioxidant properties.

DPPH and NBT Assays reported the antioxidant activities of kaempferol-7-O-rhamnoside [35]. The content had an excellent inhibitory effect on rat aldose reductase [51]. The flavonoid glycoside was proven to be the key antioxidant in a *C. osmophloeum* twig ethanolic extract [35]. These activities are associated with beneficial health effects. Therefore, *cinnamomum* has been widely used for its medicinal purpose and in nutritional food [52].

#### 3.1.4. Antidyslipidemic Activities

The activity of *C. osmophloeum* in lowering the cholesterol level was investigated by Lin et al. They showed that both Kaempferol and kaempferitrin from *C. osmophloeum* have antidyslipidemic activity [38]. Furthermore, the S-(þ)-linalool in *C. osmophloeum* was also reported to have hypolipidemic activity by inhibiting the accumulation of body fat through downregulating adipocyte differentiation [39]. Two in vivo studies disclosed encouraging results demonstrating that S-(þ)-linalool was effective at cutting lipid accumulation. They also showed that low-density lipoprotein (LDL) was significantly decreased compared to high-density lipoprotein (HDL) [38,39].

#### 3.1.5. Anti-Hyperglycemic Activities

Diabetes mellitus (DM) has become a serious medical problem. Numerous studies have been conducted by adapting natural materials as DM therapy. Among these bio-resource candidates, *C. osmophloeum* is one of the promising candidates. Proanthocyanidin and tannin from *C. osmophloeum* may possess anti-hyperglycemic activity [21]. CoTE (*C. osmophloeum* twig extracts) showed protein tyrosine phosphatase 1B inhibitory (PTP1B) activity to improve the insulin or leptin resistance [18]. Linalool, the constituent of the essential oil of *C. Osmophloeum*, might ameliorate the pancreatic values of thiobarbituric acid reactive substances. The activities of superoxide dismutase and glutathione reductase ameliorated pancreatic levels of IL-1β and NO in vivo in streptozotocin-diabetes mellitus (STZ-DM)) animals [3]. It was reported that a higher degree of polymerization of proanthocyanidins correlates with the inhibitory activity of α-glucosidase and α-amylase [21]. The studies explored leaves and twigs with different contents for decreasing the blood glucose level and treating diabetes mellitus. The results indicated that the leaves were more effective than the twigs, which contained less linalool. Moreover, in a practical sense, the leaves were much easier to preserve than twigs.

#### 3.1.6. Effects on the Cardiovascular System

Cinnamaldehyde was shown to be one of the protective ingredients for cerebral microvascular endothelial cells in mice induced by aortic banding (AB) by reducing IL-1β-induced cyclooxygenase-2 (COX-2) activity [53]. In addition, the protective role of cinnamaldehyde in cardiac hypertrophy induced by AB was related to its regulatory effect on the ERK1/2 signaling pathway [7]. The effect on the cardiovascular system was related to the anti-inflammatory mechanism [7,53]. The authors introduced cinnamaldehyde as a major compound extracted from *C. osmophloeum* and it was bought from Sigma-Aldrich (USA). It was discussed that *Cinnamomum cassia* (*C. cassia*) would be a potential drug for cardiovascular disease [7]. Unreliable cinnamon species (*C. cassia* or *C. osmophloeum*) used in the report would be a critical issue to clarify because the authors may have been inconsistent in discussing the species of cinnamon.

#### 3.1.7. Effect on Renal Disease and Anti-Hyperuricemia

It was reported that cinnamaldehyde in *C. osmophloeum* successfully inhibited the high glucose-induced hypertrophy of renal interstitial fibroblasts, as indicated by the decreased cell size; the falling cellular hypertrophy index; and the descending protein levels of collagen IV, fibronectin, and α-smooth muscle actin [4]. In order to decrease the level of uricemia, several in vivo study reports confirmed the proposed mechanisms [2,5]. The cinnamaldehyde type of *C. osmophloeum* leaf oil demonstrated anti-hyperuricemia effects through its xanthine oxidase inhibitory activity. Xanthine oxidase can catalyze the oxidation of hypoxanthine/xanthine to produce uric acid. Gout and hyperuricemia are caused by an excessive accumulation of uric acid [2,19].

#### 3.1.8. Anti-Tumor and Anti-Cancer Activity

Trans-cinnamaldehyde, a bioactive content of *C. osmophloeum*, showed an inhibition of tumor growth [46,54,55]. It could stimulate Ca^2+^ entry with subsequent cell membrane scrambling and cell shrinkage, hallmarks of eryptosis, and the suicidal death of erythrocytes [46]. Trans-cinnamaldehyde exposure induced cell death via caspase-dependent and -independent pathways resulting from the depletion or induction of ROS [54]. Lignan ester, one of the *C. osmophloeum* constituents, was shown to be a possible anticancer compound (liver and oral cancer). The cytotoxicity had a significant effect on human liver cancer (HepG2 and Hep3B) and oral cancer (Ca9-22) cells [42].

### 3.2. Potential Use of C. osmophloeum for the Treatment of Oral Mucositis (OM)

The current protocols of medicine for chemotherapy are associated with oral mucositis. Cytarabine, high-dose 5-fluororacil, alkylating agents, and platinum-based compounds are highly associated with the incidence of oral mucositis. Actinomycin D, amsacrin, bleomycin, busulfan, capecitabine, carboplatin, chlorambucil, cisplatin, cytarabine, docetaxel, doxorubicin, etoposide, floxuridine, ifofsamide, irinotecan, leucovorin, methotrexate, mitoxantron, oxaliplatin, paclitaxel, plicamycin, tioguanin, vinblastine, vincristine, vindecine, and vinorelbine (the protocol can be combined) are the medicines used for chemotherapy and are reported to lead to the development of oral mucositis [56,57]. In order to prevent oral mucositis, the Multinational Association of Supportive Care in Cancer (MASCC) and International Society of Oral Oncology (ISOO) help clinics by comprising Clinical Practice Guidelines for Oral Mucositis [58]. The recommendations are comprised of basic oral care, growth factors and cytokines, anti-inflammatory agents, laser and other light therapy, cryotherapy, and natural and miscellaneous agents (Table 3). Köstler et al. provided several experimental approaches to treat oral mucositis; they included locally applied non-pharmacological methods, anti-inflammatory and mucosa protectant agents, cytokines, granulocyte colony-stimulating factor (G-CSF, filgrastim) and granulocyte-macrophage colony-stimulating factor (GM-CSF, molgramostim), antiseptic agents, corticosteroids, mouth-coating agents, and dexpanthenol [57].

The study of *C. osmophloeum* and/or its constituents in OM are limited. In one study, the effect of cinnamaldehyde on oral mucositis and an evaluation of the salivary total antioxidant capacity of gamma-irradiated rats were carried out. The saliva samples were taken from the rats in triplicate [59]. In order to evaluate the consequences and severity of mucositis, the conditions of the oral cavity were assessed by using Parkin’s clinical scale, where 0 represents normal mucosa, 0.5 indicates normally pink mucosa, 1 stands for minor red mucosa, 2 is severe red mucosa, 3 is local desquamation, 4 describes exudation and crust around less than half of the lip area, and 5 characterizes exudation and crust for more than half of the lip area [60]. The authors concluded that the clinical effects in the intervention group seemed to be due to the antioxidant, antibacterial, and anti-inflammatory effects of cinnamaldehyde. It is noteworthy to mention that through anti-inflammation and antioxidant mechanisms, cinnamaldehyde would delay the onset of oral mucositis. Moreover, alteration in the oral microflora of existing bacteria in the fourth phase (ulceration phase) could exacerbate the severity of mucositis, whereas cinnamaldehyde alleviated oral mucositis via its antibacterial properties [59]. A recent investigation reported the effect of cinnamon bark fractions (an essential oil and an aqueous extract) on *Candida albicans* growth inhibition (growth, biofilm formation, and adherence properties) and oral epithelial cells (barrier integrity and inflammatory response) [61]. The anti-adherence and anti-inflammatory properties of proanthocyanidins, a family of polyphenols containing flavan-3-ol oligomers and polymers, are used to treat oral infections [61,62]. The two pro-inflammatory cytokines, IL-6 and IL-8, which serve as important cytokines in the development of oral mucositis, were reduced by an aqueous extract enriched with proanthocyanidins of the cinnamon fraction. This shows that the cinnamon presented in the study may be a promising agent in the alleviation of oral mucositis [61].

Other biomaterials or herbal products, such as *Aloe vera*, *Acacia catechu*, Chamomile, Hangeshashinto, indigowood root (*Isatis indigotica Fort.*), honey, *Traumeel S*, water grass decoction, and *Weleda Ratanhia*, also showed similar effects on the alleviation of OM via anti-inflammation activities [17,63,64,65,66,67,68,69,70,71,72]. The bioactive properties of the yarrow plant (*Achillea mileofolium*), honey, *Callendula officinalis* flowers, *Hipophae rhamnoides L.* plant, Chamomile, and *Aloe vera* were associated with antioxidant, anti-inflammatory, antibacterial, and wound healing effects of oral mucositis therapy (Table 4) [64,65,66,72,73,74,75,76,77]. Similar to the advantages of the medical-biological activities of these herbal agents, *C. osmophloeum* would provide an identical mechanism for relieving oral mucositis.

According to antibacterial mechanisms, several studies demonstrated bacterial changes due to radiotherapy and/or chemotherapy [85]. The bacteria were *Gemella haemolysans*, *Streptococcus mitis* [86], *Escherichia coli*, *Pseudomonas aeruginosa*, *Enterobacter* sp., *Klebsiella pneumonia* [87], *Staphylococcus aureus*, *Staphylococcus epidermidis*, *Parvimonasmicra*, *Fusobacterium nucleatum*, *Treponema denticola*, *C. glabrata*, *C. kefyr* [88], and *Porphyromonas gingivalis* [89]. Among these bacteria, *C. osmophloeum* has been confirmed to possess the antibacterial properties of *Pseudomonas aeruginosa*, *Staphylococcus aureus*, and *Staphylococcus epidermidis* (Figure 3) [32].

Flavanoid-rich fractions containing kaempferitin in *Bauhinia forficata* leaves have been investigated and shown to be effective ingredients for preventing the intestinal toxic effects of irinotecan chemotherapy. It was stated that kaempferitin, as one of the major active contents of *C. osmophloeum*, has been tested to prevent or reduce the severity of intestinal mucositis [26,90]. The chemotherapy drug 5-Fluororacil possesses side effects (i.e., induces mucositis manifestations in oral and gastrointestinal after chemotherapy). *Chimonatus Nitens var. salicifolius* aqueous extract contains three flavonoid contents: quercetin, kaempferol, and rutin, which might have an anti-inflammatory effect on gastrointestinal mucositis [91]. Kaempferol, cinamic acid, and nine other constituents obtained in mucoadhesive propolis agent have been proven to be effective in reducing radiation-induced oral mucositis. A clinical study of 24 patients revealed that mucositis only developed in two patients and each developed grade 1 mucositis and grade 2 mucositis, respectively; however, in the remaining 20 patients, mucositis did not develop [92].

Previous studies investigated the potential of *C. osmophloeum* to reduce oral mucositis. Nonetheless, *C. osmopohloeum*, which is a species of cinnamon, faces challenges in its use as an oral treatment. In addition, this review has several limitations. First, there are limited data on the medical-biological effects of *C. osmophloeum* and its potential use in oral mucositis therapy. Secondly, the reported events related to oral stomatitis allergy induced by cinnamon should be a concern [93,94,95]. In summary, *C. osmophloeum* and its constituent are anticipated to be effective and efficient in reducing the severity of OM by preventing secondary bacterial infection through their bactericidal activity, preventing the development of the second phase of OM (the primary damage response) or interrupting the third phase in which pro-inflammatory cytokines could enhance and accelerate the process of wound healing (Figure 4).

## 4. Conclusions

In conclusion, in vitro and animal studies have revealed the various medical-biological activities of *C. osmophloeum* encompassing anti-inflammatory, antibacterial, antioxidant, anti-tumor and anti-cancer, anti-hyperglycemic, anti-hyperuricemia, and antidyslipidemic activities and the effect on the cardiovascular system that have been used for renal disease therapy. Various chemical contents identified in *C. osmophloeum* contribute to its valuable medical bioactivities. Several medical-biological activities are attributed to the active ingredients of *C. osmophloeum* for OM therapy. The key to OM alleviation could be the anti-inflammatory effect of *C. osmophloeum*, as the activation of several transcription factors might release the inflammatory reaction in the second phase of OM. The antibacterial and antioxidant properties of CO might also alleviate OM. These effects might occur during the first phase of OM, preventing the generation of reactive oxygen species, or in the fourth phase, inhibiting colonized bacteria. It was suggested that bacterial cell wall products could be prevented from entering the oral submucosa.

## Figures and Tables

**Figure 1 biomedicines-08-00003-f001:**
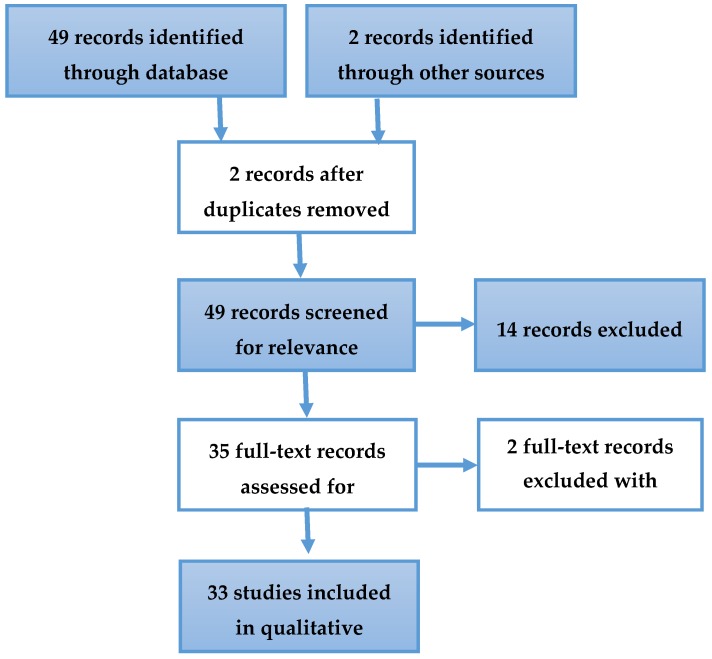
The systematic search for medical-biological activities of *Cinnamomum osmophloeum*.

**Figure 2 biomedicines-08-00003-f002:**
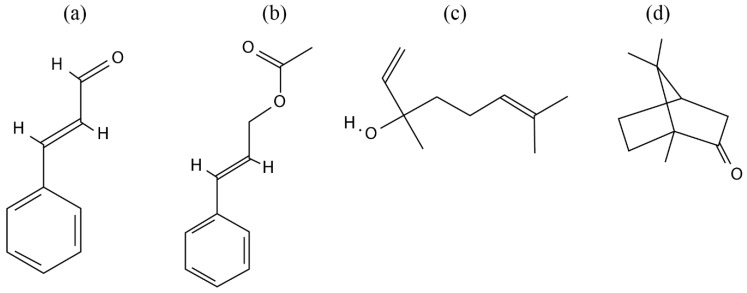
Chemical structures of *Cinnamomum osmophloeum* chemotypes: (**a**) cinnamaldehyde, (**b**) cinnamyl acetate, (**c**) linalool, and (**d**) camphor.

**Figure 3 biomedicines-08-00003-f003:**
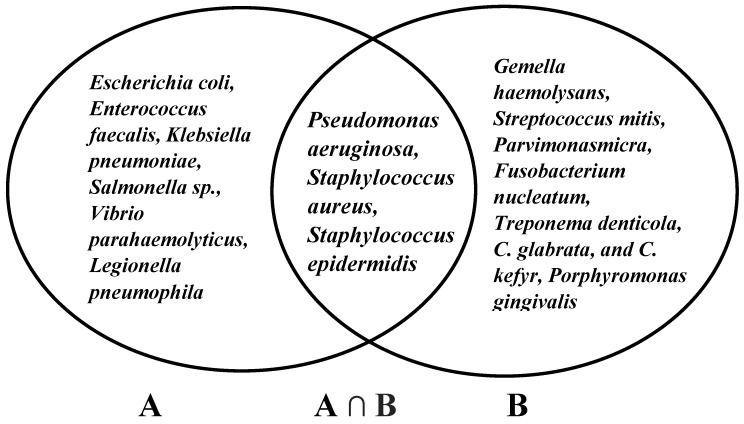
Antibacterial properties of *C. osmophloeum* (**A**) and bacterial infection of oral mucositis (**B**).

**Figure 4 biomedicines-08-00003-f004:**
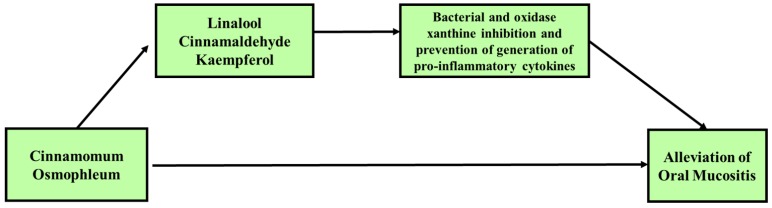
Proposed potential use of *C. osmophloeum* in the alleviation of oral mucositis (OM)**.**
*C. osmophloeum* ameliorated oxidative stress and pro-inflammation through its constituents. Several studies have investigated the anti-inflammation, antibacterial, and antioxidant activities. The review of medical-biological activities showed that *C. osmophloeum* and its constituents inhibit the pro-inflammatory response. *C. osmophloeum* potentially prevents the second phase of oral mucositis (the primary damage response) or may intercept in the third phase, with pro-inflammatory cytokines providing a positive reaction to enhance and accelerate the process of wound healing. *C. osmophloeum* was also confirmed to be bactericidal and inhibit bacteria which can reduce the severity of oral mucositis or secondary bacterial infection.

**Table 1 biomedicines-08-00003-t001:** Beneficial biological activities of *C. osmophloeum.*

Bioactivity	Chemical Identification	*C. osmophloeum* Parts	Constituent (s)	Study	Mechanisms	Reference
Anti-inflammatory effect	LC-MS/MS	Leaves	Kaempferitrin	In vitro	Down-regulate the extracellular LDL-R (chronic inflammation-related diabetes mellitus)	Ku et al., (2017) [26]
	GC-MS	Twigs	Trans-cinnamaldehyde, caryophyllene oxide, L-borneol, L-bornyl acetate, eugenol, β-caryophyllene, E-nerolidol, and cinnamyl	In vitro	Suppressing nitric oxide synthesis by LPS-stimulated macrophages	Tung et al. (2008) [30]
	GC-MS	Leaves	Trans-cinnamaldehyde,(-)-aromadendrene, caryophyllene oxide, T-cadinol, and α-cadinol	In vitro	Suppressing nitric oxide production by LPS-stimulated macrophages	Tung et al. (2010) [29]
	GC-MS and HPLC	Leaves	Cinnamaldehyde	In vitro	Cinnamaldehyde inhibits LPS-mediated pro-inflammatory cytokine production	Chao et al. (2008) [23]
	TLC	Leaves	NA	In vitro	Inhibition of the production of NO and cytokines (TNF-a and IL-12), from LPS/IFNc-activated macrophages	Fang, Rao & Tzeng (2005) [25]
	CC, HPLC, TLC, ESIMS, and GC-MS	Twigs	Kaempferol glycosides	In vitro	Nitric oxide production inhibitory activities	Lin & Chang(2012) [28]
	GC-MS	Leaves	Linalool and cinnamaldehyde	In vivo	Inhibition of the expression of molecules in both TLR4 and NLRP3 signaling pathways	Lee et al. (2018) [40]
	GC-MS	Leaves	21 compounds were identified	In vitro	Inhibition of IL-1â and IL-6 production	Chao et al. (2005) [24]
Antibacterial activity	GC	Leaves	Cinnamaldehyde	In vitro	Bactericidal	Chang, Chen & Chang (2001) [32]
	GC-MS	Leaves	Cinnamaldehyde, cinnamic acid, cinnamyl alcohol, and cinnamyl acetate	In vitro	Bacterial inhibition	Chang et al. (2008) [31]
Antifungal activity	GC-MS	Leaves	Cinnamaldehyde	In vitro	NA	Cheng et al. (2006) [34]
	GC-MS	Leaves	Cinnamaldehyde	In vitro	NA	Wang, Chen & Chang (2005) [33]
Antioxidant activities	ESIMS	Twigs	Kaempferol-7-O-rhamnoside	In vitro	NA	Chua, Tung, & Chang (2008) [35]
	GC-MS and GC−FID	Leaves	Alloaromadendrene	In vitro	NA	Yu et al. (2014) [41]
	GC-MS and GC-FID	Leaves	Trans-cinnamaldehyde	In vitro	NA	Yeh et al. (2013) [37]
	GC-MS and GC-FID	Leaves	Trans cinnamaldehyde	In vivo	Expression of antioxidative-related genes was pointedly affected by essential oils from *C. osmophloeum*.	Hsu et al. (2012) [36]
Antidyslipidemic activity	HPLC	Leaves	Kaempferol and kaempferitrin	In vivo	Cholesterol-lowering activity	Lin et al. (2011) [38]
Anti-hyperglycemic and antioxidant activities	A modified vanillin-H_2_SO_4_ assay A modified acid-butanol assay The AlCl_3_ method	Twigs	Proanthocyanidin and tannin contents	In vitro	CoTE has PTP1B inhibitory activity to improve insulin or leptin resistance	Lin et al. (2016) [18]
Hepatoprotective effects	NA	Leaves	trans-cinnamaldehyde, ()-aromadendrene, T-cadinol, or R-cadinol	In vivo	The modulation of anti-inflammatory activities (decreased the aspartate aminotransferase (AST), alanine aminotransferase (ALT), tumor necrosis factor-R (TNF-R), and interleukin 6 (IL-6) levels in serum)	Tung et al. (2011) [6]
Pancreas Protective Effect and Hypoglycemic activity	GC/MS	Leaves	Linalool	In vivo	1. Decreased pancreatic values of thiobarbituric acid reactive substances and activities of superoxide dismutase and glutathione reductase 2. Decreased pancreatic levels of interleukin-1β and nitric oxide	Lee et al. (2013) [3]
Prevent Cardiac Hypertrophy	HPLC	Leaves	Cinnamaldehyde	In vivo	The protective role of cinnamaldehyde related to the ERK1/2 signaling pathway.	Yang et al. (2015) [7]
Treatment of renal interstitial fibroblasts	NA	Leaves	Cinnamaldehyde	In vitro	Inhibit high glucose-induced hypertrophy (decreased cell size; cellular hypertrophy index; and protein levels of collagen IV, fibronectin, and α-smooth muscle actin).	Chao et al. (2010) [4]
Anticancer (liver and oral cancer)	TLC, CC and HPLC	Heart wood and roots	Lignan Esters	In vitro	Tumor cell growth inhibition	Chen et al. (2010) [42]
Anti-diabetes	TLC	Twigs	Kaempferol glycosides CO-1 and CO-2	In vitro	Enhanced adiponectin secretion, and activation of the insulin signaling pathway	Lee et al. (2009) [43]
Anti-hyperuricemia effect	GC-MS	Leaves	Cinnamaldehyde	In vivo	Acts as a xanthine oxidase inhibitor and reduces the serum uric acid levels	Wang et al. (2008) [2]
Anxiolytic properties	HPLC	Leaves	Linalool	In vivo	Reduced the amount of 5-HT, DA and NE and increased the level of dopamine in striatum	Cheng et al. (2014) [44]
Wound Repair Promoter and Antioxidant	NA	Leaves	NA	In vitro and in vivo	Inhibited tyrosinase activity and reduced melanin content	Lee et al. (2015) [20]
Anti-inflammatory and anti-cancer properties	NA	Barks	NA	In vivo	The growth inhibition of NO, TNF-, and IL-12, and tumor cell proliferation	Rao et al. (2007) [22]
Hypolipidemic effects	NA	Leaves	S-(þ)-linalool	In vivo	Inhibited lipid accumulation through downregulation of 3T3-L1 adipocyte differentiation	Cheng et al. (2018) [39]
Effect on the human immune system	HS-GC/MS and HPLC	Leaves	Cinnamaldehyde	In vivo	Cytokines modulatory effect	Lin et al. (2011) [45]
Potential skin-whitening and protective agent	NA	Leaves	Cinnamaldehyde and cinnamylacetate	In vitro	Neutralized the IBMX-induced increase in melanin content in B16-F10 cells by inhibiting tyrosinase gene expression at the level of transcription	Lee et al. (2015) [20]
Anti-inflammatory effect in intestine	GC/MS	Leaves	Linalool	In vivo	The suppression of the TLR4 pathway by CO and partly by the inhibitory effect of CO on the activity of xanthine oxidase	Lee et al. (2015) [27]
Anti-tumor	NA	Leaves	Trans-cinnamaldehyde	In vitro	Trans-cinnamaldehyde triggers suicidal death oferythrocytes, i.e., cells devoid of mitochondria and gene expression.	Theurer et al. (2013) [46]
Dietary supplements and treatment of hyperuricemia and gout	GC-MS and GC-FID	Leaves	Cinnamaldehyde	In vitro	The xanthine oxidase inhibitory activity	Huang et al. (2018) [19]
Anti-hyperglycemic and antioxidant activities	(MALDI/MS) (RP-HPLC) /MS/MS	Twigs	Proanthocyanidin	In vitro	The proanthocyanidins in CoTE mainly consisted of (epi)catechin and contained at least one A-type linkage. The inhibitory activity of α-glucosidase and α-amylase	Lin et al. (2016) [21]

NA: Not available; LC: liquid chromatography; MS: mass spectrometry; GC: gas chromatography; HPLC: high-performance liquid chromatography; TLC: thin layer chromatography; CC: column chromatography; ESIMS: electrospray ionization mass spectrometry; GC-FID: gas chromatography−flame ionization detection; MALDI: matrix-assisted laser desorption/ionization; RP: reverse phase.

**Table 2 biomedicines-08-00003-t002:** Chemotypes and chemical compositions of *C. osmophloeum*. All of the studies on *C. osmophloeum* chemical contents utilized leaf extracts, except number 48 (twig extract).

No	Chemical Compositions	References
**Cinnamaldehyde type**		
1	Cinnamaldehyde, Geranyl acetate, Benzaldehyde.	Chang et al. (2001), Wang et al. (2005) [32,33]
2	*trans*-Cinnamaldehyde, Benzenepropanal, 4-Allylanisole.	Chang et al. (2008) [31]
3	*trans*-Cinnamaldehyde, Cinnamyl acetate, 3-Phenylpropionaldehyde.	Tung et al. (2010) [29]
4	*trans*-Cinnamaldehyde, Cinnamyl acetate, 3-Phenylpropionaldehyde.	Tung et al. (2010) [29]
5	*trans*-Cinnamaldehyde, Cinnamyl acetate, 3-Phenylpropionaldehyde.	Tung et al. (2010) [29]
6	*trans*-Cinnamaldehyde, Cinnamyl acetate, 3-Pheaylpionaldehyde.	Wang et al. (2008) [2]
7	*trans*-Cinnamaldehyde, *trans*-Cinnamyl acetate, 3-Phenylpropionaldehyde.	Hsu et al. (2012) [36]
8	*trans*-Cinnamaldehyde, Benzenepropanal, Benzaldehyde.	Yeh et al. (2013), Cheng (2008) [37,49]
9	*trans*-Cinnamaldehyde, Benzaldehyde, Benzenepropanal.	Yeh et al. (2013) [37]
10	*trans*-Cinnamaldehyde, Benzaldehyde, Benzenepropanal.	Yeh et al. (2013) [37]
11	*trans*-Cinnamaldehyde, Benzaldehyde, Benzenepropanal.	Yeh et al. (2013) [37]
12	*trans*-Cinnamaldehyde, Linalool, *trans*-Cinnamyl acetate.	Yeh et al. (2013) [37]
13	*trans*-Cinnamaldehyde, Benzaldehyde, 3-Phenylpropionaldehyde.	Huang et al. (2018) [19]
14	*trans*-Cinnamaldehyde, *trans*-Cinnamyl acetate, Benzenepropanal.	Mdoe et al. (2014) [47]
15	Cinnamaldehyde, Geranyl acetate, Benzaldehyde.	Wang et al. (2005) [33]
16	*trans*-Cinnamaldehyde, Cinnamyl acetate, β-Caryophyllene.	Cheng et al. (2004) [48]
17	*trans*-Cinnamaldehyde, Benzenepropanal, benzaldehyde.	Cheng et al. (2004) [48]
18	*trans*-Cinnamaldehyde, Benzenepropanal, benzaldehyde.	Cheng et al. (2004) [48]
19	*trans*-Cinnamaldehyde, Benzenepropanal, 4-Allylanisole.	Cheng et al. (2009) [50]
20	*trans*-Cinnamaldehyde, Cinnamyl acetate, Bornyl acetate.	Cheng et al. (2006) [34]
21	*trans*-Cinnamaldehyde, Benzenepropanal, Eugenol.	Cheng et al. (2006) [34]
22	*trans*-Cinnamaldehyde, Benzenepropanal, Cinnamyl acetate.	Cheng et al. (2006) [34]
**Cinnamaldehyde/Cinnamyl acetate type**		
23	Cinnamyl acetate, *trans*-Cinnamaldehyde, Benzenepropanal.	Chang et al. (2008) [31]
24	*trans*-Cinnamaldehyde, Cinnamyl acetate, Benzenepropanal.	Cheng et al. (2004) [48]
25	Cinnamyl acetate, *trans*-Cinnamaldehyde, Camphene.	Cheng et al. (2004) [48]
26	Cinnamyl acetate, *trans*-Cinnamaldehyde, Benzenepropanal.	Cheng et al. (2009) [50]
27	*trans*-Cinnamaldehyde, Cinnamyl acetate, Benzenepropanal.	Cheng et al. (2006) [34]
28	Cinnamyl acetate, *trans*-Cinnamaldehyde, Camphene.	Cheng et al. (2006) [34]
**Cinnamyl acetate type**		
29	Cinnamyl acetate, 2-Methylbenzofuran, Geranyl acetate.	Chang et al. (2008) [31]
30	Cinnamyl acetate, *trans*-Cinnamaldehyde, Caryophyllene oxide.	Tung et al. (2010) [29]
31	Cinnamyl acetate, 2-Methylbenzofuran, Geranyl acetate.	Cheng et al. (2009) [50]
32	Cinnamyl acetate, 2-Methylbenzofuran, Geranyl acetate.	Cheng et al. (2006) [34]
**Linalool type**		
33	Linalool, *trans*-Cinnamaldehyde, Cinnamyl acetate.	Chang et al. (2008) [31]
34	Linalool, β-Caryophyllene, 4-Allylanisole.	Tung et al. (2010) [29]
35	Linalool, Cinnamaldehyde, 3-phenyl-2-propenal.	Lee et al. (2013) [3]
36	Linalool, Citral, Coumarin.	Cheng et al. (2004) [48]
37	Linalool, *trans*-Cinnamaldehyde, 4-Allylanisole.	Cheng et al. (2009) [50]
38	Linalool, Coumarin, *trans*-Cinnamaldehyde.	Cheng et al. (2006) [34]
**Camphor type**		
39	Camphor, L-Bornyl acetate, (+)-Limonene.	Chang et al. (2008) [31]
40	D-(+)-Camphor, L-Bornyl acetate, α-Terpineol	Tung et al. (2010) [29]
41	D-(+)-Camphor, L-Bornyl acetate, α-Terpineol.	Hsu et al. (2012) [36]
42	Camphor, bornyl acetate, Limonene.	Cheng et al. (2004) [48]
43	Camphor, L-Bornyl acetate, (+)-Limonene.	Cheng et al. (2009) [50]
44	Camphor, Bornyl acetate, Limonene.	Cheng et al. (2006) [34]
**Mixed type**		
45	Spathulenol, Linalool, α-Terpineol.	Chang et al. (2008) [31]
46	Neral, 1.8-Cineol, Linalool.	Chang et al. (2001), Wang et al. (2005) [32,33]
47	L-Bornyl acetate, Caryophyllene oxide, γ-Eudesmol.	Tung et al. (2008) [30]
48	L-Bornyl acetate, α-Cadinol, T-Cadinol.	Tung et al. (2010) [29]
49	L-bornyl acetate, T-Cadinol, α -Cadinol.	Yu et al. (2014) [41]
50	1,8-cineole, spathulenol, santolina triene.	Chao et al. (2005) [24]
51	T-Cadinol, α -Cadinol., bornyl acetate.	Cheng et al. (2004) [48]
52	Geranial, Neral, 1,8-Cineole.	Cheng et al. (2009) [50]
53	T-Cadinol, α -Cadinol, bornyl acetate.	Cheng et al. (2006) [34]

**Table 3 biomedicines-08-00003-t003:** Interventions to prevent oral mucositis (clinical practice guidelines) proposed by the Multinational Association of Supportive Care in Cancer (MASCC) and International Society of Oral Oncology (ISOO).

Intervention	Protocol	Population	Evidence of Effectiveness
Basic oral care	Tooth brushing, flossing, and one mouth rinse	All age groups and across all cancer treatment modalities	Not strong evidence
Growth factors and cytokines	Palifermin (keratinocyte growth factor-1)	Patients receiving high-dose chemotherapy and total body irradiation, followed by autologous stem cell transplantation for hematological malignancies	Strong evidence
Anti-inflammatory agents	Benzydamine mouthwash	Patients with head and neck cancer receiving moderate-dose radiation therapy (up to 50 Grays), without concomitant chemotherapy	Strong evidence
Laser and other light therapy	Low-level laser therapy (LLLT)	Patients receiving high-dose chemotherapy for HSCT with or without total body irradiation	Strong evidence
Cryotherapy	The placement of ice chips in the mouth	Patients receiving bolus dosing of 5-fluorouracil	Strong evidence
Natural and miscellaneous agents	Systemic zinc supplements administered orally (antioxidant effect)	Patients with oral cancer undergoing radiotherapy or chemoradiation	Not strong evidence

**Table 4 biomedicines-08-00003-t004:** Bioactive properties of natural agents for oral mucositis therapy.

Natural Agents	Bioactivity	References
Yarrow Plant (*Achillea millefolium*)	Anti-bacterial and anti-inflammatory effect	Mirazandeh et al. (2014) [73]
Manuka Honey (*Leptospermum scoparium*)	Wound healing and anti-microbial	Hawley et al. (2013) [72]
Weleda Pflanzen-Zahngel and Weleda Ratanhia-Mundwasser	Anti-inflammatory, anti-bacterial, and lesion healing	Tiemann et al. (2007) [70]
*Calendula officinalis* flowers	Anti-inflammatory, anti-bacterial, and anti-oxidant	Babaee et al. (2013) [74]
Honey and coffee	Antioxidant, anti-microbial, and anti-inflammatory	Raeessi et al. (2014) [63]
*Aloe vera*	Anti-inflammatory, bactericidal, and wound healing	Sahebjamee et al. (2015) [71]
Hangeshashinto: *Pinelliae tuber*, *Scutellariae Radix*, *Glycyrrhizae Radix*, *Zizyphi Fructus*, *Ginseng Radix*, *Zingiberis Processum rhizoma*, *and Coptidis rhizome*	Anti-inflammatory	Aoyama et al. (2014) [67]
Indigowood Root (*Isatis indigotica Fort.*)	Anti-inflammatory	You et al. (2009) [69]
Topical Honey	Anti-inflammatory, anti-microbial, and wound healing	Khanal et al. (2010) [75]
*Hippophae rhamnoides L.* plant	Anti-oxidant, anti-ulcerogenic, anti-inflammatory, anti-microbial, and proinflammatory cytokine Antagonist	Kuduban et al. (2016) [76]
Honey from the clover plant *Trifolium alexandrenum*	Anti-microbial	Rashad et al. (2009) [78]
Qingre Liyan decoction	Anti-oxidant and anti-inflammatory	Lambros et al. (2014) [79]
Hangeshashinto	Anti-inflammatory and anti-microbial	Kono et al. (2014) [65]
Chamomile	Anti-inflammatory, anti-bacterial, and antifungal	Fidler et al. (1996) [64]
*Rhodila algida*	Anti-oxidant and immunostimulant	Loo et al. (2010) [80]
Qingre Liyan Decoction	Enhancing body immunity and promoting salivary EGF	Wu et al. (2007) [81]
Chamomile	Anti-inflammatory, anti-bacterial, and anti-fungal	Pourdeghatkar et al. (2017) [66]
Pure Honey	Anti-bacterial and anti-inflammatory	Motallebnejad et al. (2008) [77]
*Aloe vera*	Anti-inflammatory, anti-bacterial, and anti-fungal	Puataweepong et al. (2009) [82]
*Aloe vera* and vitamin E	Antioxidant, anti-inflammatory, and healing properties	Cuba et al. (2015) [83]
Traumeel S	Anti-inflammatory	Sencer et al. (2012) [84]
*Chamomilla recutita*	Anti-inflammatory	Braga et al. (2015) [85]
Wild chamomile *(Matricaria recutita L.)*	Anti-inflammatory, anti-bacterial, and anti-fungal	Mazokopakis et al. (2003) [65]

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
