# Peer review of "Beneficial Biological Activities of Cinnamomum osmophloeum and Its Potential Use in the Alleviation of Oral Mucositis: A Systematic Review"

_biomedicines, 2020, doi:10.3390/biomedicines8010003_

Round 1
Reviewer 1 Report
Bakar et. al have conducted a systematic review of current literature exploring the various biological activities of Cinnamomum Osmophloeum in altering cellular pathophysiology and they propose that it may protect against oral mucositis (OM) in cancer patients receiving chemo or radiotherapy. The authors have also compared the biological activities of C. Osmophloeum with other plant products that are also being studied and used in the context of OM and suggest that since the biological activities are similar for these plat-derived products, C. Osmophloeum might be a promising option to alleviate OM. Though the study has a good design and rationale, there are several major issues that the authors should address
Page 2, lines 12-13: “The risk of oral mucositis developed in more than 50% and 8 % newly diagnosed cancer patients (112,000 patients) in United States. patients received radiotherapy in head and neck area”. What do the authors mean by this sentence? Are 50% of patients receiving chemo affected by OM? Page 2, lines 17-31: Paragraph explaining the various phases involved in OM progression is not clear. There are several ambiguous phrases and sentences and it is not clear how one phase leads to the next phase. Please rephrase this paragraph for clarity. The authors have reviewed just 32 studies for this review, which might not be sufficient to propose that C. Osmophloeum may protect against oral mucositis. Page 3: Numbers mentioned in fig 1 does not exactly match up with text in the paragraph (above fig 1) describing how literature search was done. Page 12, lines 8-9: “The investigations on antibacterial activities of Osmophloeum were limited”. What does this mean, when the authors mention in the beginning of the paragraph that cinnamon has been investigated for its anti-bacterial properties against several types of bacteria? Section 1.3. Antioxidant activities: The authors state that several studies have looked at anti-oxidative properties of C. Osmophloeum by using several assays. What are the mechanisms proposed in these studies that explain the anti-oxidative properties? Section 1.3. Antioxidant activities: Please include references for the two in vivo studies that are being described in this section. Page 12, lines 37-38: “Proanthocyanidin and tannin from Osmophloeum may have possessed the anti-hyperglycemic activity”. Please include references for the above statement. Page 13, lines 17-18: “In order to decrease the level of uricemia, several in vivo study reports have confirmed the proposed mechanisms”. Can the authors explain the mechanisms proposed in the referenced papers? Page 13, lines 20-22: “Trans-cinnamaldehyde could stimulate the Ca2+ entry with subsequent cell membrane scrambling and cell shrinkage, hallmarks of eryptosis, the suicidal death of erythrocytes [45], It might inhibit the tumour growth.” How does killing of erythrocytes result in inhibition of tumor growth? Is this a finding from the referenced paper or is this the authors’ hypothesis? Section 3.1.8. Anti-tumor and anti-cancer: How does C. Osmophloeum work as an anti-tumor or anti-cancer agent? What are the mechanisms proposed by the referenced studies? Please include the following citations and also explain their findings: i. Veilleux, M., Grenier, D. Determination of the effects of cinnamon bark fractions on Candida albicansand oral epithelial cells. BMC Complement Altern Med 19, 303 (2019) doi:10.1186/s12906-019-2730-2 ii. Oral Mucositis Complicating Chemotherapy and/or Radiotherapy: Options for Prevention and Treatment Wolfgang J. Köstler, MD; Michael Hejna, MD; Catharina Wenzel, MD; and Christoph C. Zielinski, MD. CA Cancer J Clin 2001; 51: 290-315 There are published studies showing that consuming or chewing cinnamon-flavored agents result in contact stomatitis. What are the authors’ thoughts on how this might be an issue and might potentially aggravate the patient’s mucositis? Do the authors foresee any limitations in using Osmophloeum to provide relief from OM? In addition to bacterial infections, oral mucositis may also be exacerbated by fungal infections. Does cinnamon exhibit any anti-fungal properties? Page 14: Lines 11-13: The authors state that in Ref #49, Osmophloeum has beneficial effects against oral mucositis through its anti-oxidant, anti-inflammatory and anti-bacterial properties. What are the proposed mechanisms by which C. Osmophloeum mediates protection? Please include page numbers. There are major issues with English grammar, phrasing and sentence formation that needs to be rectified.
Author Response
Comments to authors for Biomedicines manuscript:
#Reviewer 1
Bakar et. al have conducted a systematic review of current literature exploring the various biological activities of Cinnamomum Osmophloeum in altering cellular pathophysiology and they propose that it may protect against oral mucositis (OM) in cancer patients receiving chemo or radiotherapy. The authors have also compared the biological activities of C. Osmophloeum with other plant products that are also being studied and used in the context of OM and suggest that since the biological activities are similar for these plat-derived products, C. Osmophloeum might be a promising option to alleviate OM. Though the study has a good design and rationale, there are several major issues that the authors should address
Thank you! We found your valuable feedback on my manuscript and changes have been made accordingly.
Page 2, lines 12-13: “The risk of oral mucositis developed in more than 50% and 8 % newly diagnosed cancer patients (112,000 patients) in United States. patients received radiotherapy in head and neck area”. What do the authors mean by this sentence? Are 50% of patients receiving chemo affected by OM?
Thank you for your suggestion. We have rephrase the above sentence into the following sentence.
“Patients receiving radiotherapy to head and neck area had significant risk to develop oral mucositis”
Page 2, lines 17-31: Paragraph explaining the various phases involved in OM progression is not clear. There are several ambiguous phrases and sentences and it is not clear how one phase
leads to the next phase. Please rephrase this paragraph for clarity.
Thank you for catching some ambiguous phrases. We have followed your suggestion by adding some explanations (the underlined sentences) in each phases of OM development.
The first phase (initiation stage) happens after exposure to radiotherapy or chemotherapy. It consists of two events: DNA breakdown; and the generation of reactive oxygen species (ROS). DNA strand breakdowns lead to direct injury and death of the cells and reactive oxygen species play the role as key initiators and mediators of downstream biological events. During the second phase (primary damage response), activator transduction pathways are stimulated by the DNA breaks strand that could lead to the activation of several transcription factors, including p53 and nuclear factor kappa-B (NF-κB). NF-κB works as a controller for the expression of a broad range of genes, which produces a series of mediators, including pro-inflammatory cytokines and both pro- and anti-apoptotic cellular changes. During phase III (signal amplification stage), pro-inflammatory cytokines deliver a positive reaction to enhance and accelerate the process of wound healing. During phase IV (ulceration phase, also called symptomatic phase), it is common for the mucosal surface to become re-infectedwith bacteria. Bacterial invasion stimulates macrophages accumulation to conceal additional amounts of pro-inflammatory cytokines. During phase V (healing stage), signals from the connective tissue to the bordering epithelium could activate the migration, propagation, and differentiation of cells to results in healed mucosa[14,16,17].
The authors have reviewed just 32 studies for this review, which might not be sufficient to
propose that C. Osmophloeum may protect against oral mucositis.
Thank you for your feedback. C. osmophloeum is the species of cinnamon which possessed some identical chemical contents with the other species of cinnamon that have been worldwide studied by researchers. We also provide the chemical contents of C. osmophloeum to ensure the mechanism. However, related to your next suggestion we have underlined the limitation of number in our review.
. In addition, this review has several limitations. First, there are limited data on the medical-biological effects of C. osmophloeum and its potential use in oral mucositis therapy.
Page 3: Numbers mentioned in fig 1 does not exactly match up with text in the paragraph (above
fig 1) describing how literature search was done.
Thank you for catching this. We have changed the wrong paper number in the fig 1.
Page 12, lines 8-9: “The investigations on antibacterial activities of C. Osmophloeum were
limited”. What does this mean, when the authors mention in the beginning of the paragraph that
cinnamon has been investigated for its anti-bacterial properties against several types of bacteria?
Thank you for pointing this out. We have erased the confusing sentence.
Section 3.1.3. Antioxidant activities: The authors state that several studies have looked at antioxidative properties of C. Osmophloeum by using several assays. What are the mechanisms
proposed in these studies that explain the anti-oxidative properties?
Thank you for your valuable suggestion. We have added the mechanism related anti-oxidative properties.
DPPH and NBT Assays reported the antioxidant activities of kaempferol-7-O-rhamnoside [36]. The content had an excellent inhibitory effect on rat aldose reductase [47]. The flavonoid glycoside was proven to be the key antioxidant in C. osmophloeum twig ethanolic extract[36]. These activities are associated with beneficial health effects; therefore cinnamomum has been used widely for its medicinal purpose and in nutritional food[48]
Section 3.1.3. Antioxidant activities: Please include references for the two in vivo studies that are being described in this section.
Thank you for pointing this out. We have included references for the statement.
Page 12, lines 37-38: “Proanthocyanidin and tannin from C. Osmophloeum may have possessed
the anti-hyperglycemic activity”. Please include references for the above statement.
Thank you for your suggestion. We have cited the reference for the statement.
Page 13, lines 17-18: “In order to decrease the level of uricemia, several in vivo study reports
have confirmed the proposed mechanisms”. Can the authors explain the mechanisms proposed
in the referenced papers?
We appreciate your suggestion for adding the necessary explanation for the mechanism. We have added into the following sentence.
The cinnamaldehyde type of C. osmophloeum leaf oil demonstrated anti-hyperuricemia effects through its xanthine oxidase inhibitory activity. Xanthine oxidase can catalyzes the oxidation of hypoxanthine/xanthine to produce uric acid. Gout and hyperuricemia are caused by excessive accumulation of uric acid [2,20].
Page 13, lines 20-22: “Trans-cinnamaldehyde could stimulate the Ca2+ entry with subsequent
cell membrane scrambling and cell shrinkage, hallmarks of eryptosis, the suicidal death of
erythrocytes [45], It might inhibit the tumour growth.” How does killing of erythrocytes result in
inhibition of tumor growth? Is this a finding from the referenced paper or is this the authors’
hypothesis?
We respect your suggestion. We have added the explanation related the mechanisms. It was the finding from the referenced paper.
Trans-cinnamaldehyde, a bioactive content of C. osmophloeum showed inhibition of tumor growth [51-53]. It could stimulate Ca2+ entry with subsequent cell membrane scrambling and cell shrinkage, hallmarks of eryptosis, and suicidal death of erythrocytes [54]. Trans-cinnamaldehyde exposure induced cell death via caspase-dependent and -independent pathways resulting from depletion or induction of ROS [51].
Section 3.1.8. Anti-tumor and anti-cancer: How does C. Osmophloeum work as an anti-tumor or anti-cancer agent? What are the mechanisms proposed by the referenced studies?
Thank you for your suggestion. The mechanism was also related to the comment number 10 and we have added the explanations.
Please include the following citations and also explain their findings: Veilleux, M., Grenier, D. Determination of the effects of cinnamon bark fractions on Candida
albicans and oral epithelial cells. BMC Complement Altern Med 19, 303 (2019)
doi:10.1186/s12906-019-2730-2
Oral Mucositis Complicating Chemotherapy and/or Radiotherapy: Options for Prevention and
Treatment Wolfgang J. Köstler, MD; Michael Hejna, MD; Catharina Wenzel, MD; and
Christoph C. Zielinski, MD. CA Cancer J Clin 2001; 51: 290-315
We appreciate your recommendation to include the above reference. They are critical information which are valuable to our submitted manuscript. We have cited them in the paper.
There are published studies showing that consuming or chewing cinnamon-flavored agents result in contact stomatitis. What are the authors’ thoughts on how this might be an issue and might
potentially aggravate the patient’s mucositis?
We appreciate your feedback related the above issue.
Do the authors foresee any limitations in using C. Osmophloeum to provide relief from OM?
Yes. The limitations related to the oral allergic of cinnamon and we have added explanation in the manuscript.
Secondly,the reportedevents related to oral stomatitis allergy induced by cinnamon should be a concern
In addition to bacterial infections, oral mucositis may also be exacerbated by fungal infections.
Does cinnamon exhibit any anti-fungal properties?
Yes. We have added the antifungal properties although the properties of C. osmophloeum was applied in wood fungi. Additionally, your recommendation to cite the article of the anti-fungal properties of other species cinnamon provided the valuable information to explain the anti-fungal properties related OM.
Page 14: Lines 11-13: The authors state that in Ref #49, C. Osmophloeum has beneficial effects
against oral mucositis through its anti-oxidant, anti-inflammatory and anti-bacterial properties.
What are the proposed mechanisms by which C. Osmophloeum mediates protection?
Thank you for your feedback. We have explained the mechanisms in figure 4.
Please include page numbers.
Thank you for your suggestion. We have included page numbers.
There are major issues with English grammar, phrasing and sentence formation that needs to be rectified
Thank you for your valuable suggestion. We have followed your suggestion by proofing the language.
Reviewer 2 Report
Authors have submitted a review article titled Medical-Biological Activities of Cinnamomum Osmophloeum and its Potential Use in Alleviation of Oral Mucositis: A Systematic Review. It is quite an interesting review paper which cover details of Cinnamomum Osmophloeum and its biological activities including anti- microbial, inflammatory, oxidant, cancer, diabetes, hyperlipidemic and hyperuricemia. Focusing of review is good. However, I would like to give some suggestion as follows
Provide Cinnamomum Osmophloeum details such as plant kingdom, major distribution worldwide, whether traditionally used this plant for OM anywhere etc. in the introductory section.
Line no 12-13 the risk of oral mucositis developed in more than 50% and 8 % newly diagnosed cancer patients (112,000 patients) in the United States. It is confusing statements and changes these lines.
Line No 38 -42 and 45-46 have repeated these sentences as in abstract better to avoid these lines in abstract, write as reported articles were performed using web search engine tools in the abstract section
Antibacterial activities line No 2-5 several pathogenic bacteria, not pathogen bacteria and must do italic for all microbial name and plant names in the whole manuscript
Species name always started with small letter C. Osmophloeum is not correct, C. osmophloeum is a correct format. Please change it all in the entire manuscript
What does the meaning of in-vitro animal studies, if animals used for any experiments that can be called in-vivo
Minor language improvement is needed
I could not check the plagiarism of this manuscript.
Author Response
Comments to authors for Biomedicines manuscript:
#Reviewer 2
Authors have submitted a review article titled Medical-Biological Activities of Cinnamomum Osmophloeum and its Potential Use in Alleviation of Oral Mucositis: A Systematic Review. It is quite an interesting review paper which cover details of Cinnamomum Osmophloeum and its biological activities including anti- microbial, inflammatory, oxidant, cancer, diabetes, hyperlipidemic and hyperuricemia. Focusing of review is good. However, I would like to give some suggestion as follows
Provide Cinnamomum Osmophloeum details such as plant kingdom, major distribution worldwide, whether traditionally used this plant for OM anywhere etc. in the introductory section.
We appreciate and we agree with your idea. We have follow your suggestion by composing the following sentence.
Cinnamon plants (Cinnamomum spp.) are of the genus Lauraceae native to South and Southeast Asia and are generally used as food flavors and as traditional medicinal plants.
Line no 12-13 the risk of oral mucositis developed in more than 50% and 8 % newly diagnosed cancer patients (112,000 patients) in the United States. It is confusing statements and changes these lines.
Thank you for your valuable feedback. We have rephrase the above sentence.
Patients receiving radiotherapy to head and neck areas are at significant risk of developing oral mucositis. The risk would be lower (less than 50% or little risk) in patients withprolonged chemotherapy, patients receiving surgery, and patients with radiotherapy to non-head and neck area [10,13]
Line No 38 -42 and 45-46 have repeated these sentences as in abstract better to avoid these lines in abstract, write as reported articles were performed using web search engine tools in the abstract section
Thank you for your valuable suggestion. We have changed the lines in the abstract section.
Antibacterial activities line No 2-5 several pathogenic bacteria, not pathogen bacteria and must do italic for all microbial name and plant names in the whole manuscript
We appreciate your suggestion for pointing this out. We have changed them into the italic font.
The literature survey indicated that several pathogenic bacteria including Escherichia coli, Pseudomonas aeruginosa, Enterococcus faecalis, Staphylococcus aureus, Staphylococcus epidermidis, methicillin-resistant Staphylococcus aureus (MRSA), Klebsiella pneumoniae, Salmonella sp., and Vibrio parahaemolyticus have been used to investigate the antibacterial activity of C. osmophloeum [30].
Species name always started with small letter C. Osmophloeum is not correct, C. osmophloeum is a correct format. Please change it all in the entire manuscript
Thank you for your valuable information. We have replaced by the correct format.
What does the meaning of in-vitro animal studies, if animals used for any experiments that can be called in-vivo
Thank you for catching this ambiguous phrase. It was a typo error. The word “and” accidently missed between in-vitro and animal.
Minor language improvement is needed
Thank you for your valuable suggestion. We have followed your suggestion.
I could not check the plagiarism of this manuscript.
We respect your work for the manuscript.